# Influence of Laser Intensity Fluctuation on Single-Cesium Atom Trapping Lifetime in a 1064-nm Microscopic Optical Tweezer

**Rui Sun [1], Xin Wang [1], Kong Zhang [1], Jun He [1,2] and Junmin Wang [1,2,*]**

[1]  State Key Laboratory of Quantum Optics and Quantum Optics Devices, and Institute of Opto-Electronics, Shanxi University, Taiyuan 030006, China; 201912607009@email.sxu.edu.cn (R.S.); 201912607014@email.sxu.edu.cn (X.W.); 201712601010@email.sxu.edu.cn (K.Z.); hejun@sxu.edu.cn (J.H.)

[2]  Collaborative Innovation Center of Extreme Optics of the Ministry of Education and Shanxi Province, Shanxi University, Taiyuan 030006, China

*  Correspondence: wwjjmm@sxu.edu.cn



**Featured Application: Authors are encouraged to provide a concise description of the specific application or a potential application of the work. This section is not mandatory.**

**Abstract:** An optical tweezer composed of a strongly focused single-spatial-mode Gaussian beam of a red-detuned 1064-nm laser can confine a single-cesium (Cs) atom at the strongest point of the light intensity. We can use this for coherent manipulation of single-quantum bits and single-photon sources. The trapping lifetime of the atoms in the optical tweezers is very short due to the impact of the background atoms, the parametric heating of the optical tweezer and the residual thermal motion of the atoms. In this paper, we analyzed the influence of the background pressure, the trap frequency of optical tweezers and the laser intensity fluctuation of optical tweezers on the atomic trapping lifetime. Combined with the external feedback loop based on an acousto-optical modulator (AOM), the intensity fluctuation of the 1064-nm laser in the time domain was suppressed from ±3.360% to ±0.064%, and the suppression bandwidth in the frequency domain reached approximately 33 kHz. The trapping lifetime of a single-Cs atom in the microscopic optical tweezers was extended from 4.04 s to 6.34 s.

**Keywords:** optical tweezer; atomic trapping lifetime; parametric heating; suppression of laser intensity fluctuation

## 1. Introduction

The physical implementation of a single-photon source has important application value in the fundamental research of quantum optics and linear quantum computation, especially the controllable triggered single-photon source as its core quantum source. Generally, one can generate single-photons via single-atom, single-molecule, single-ion, single-quantum dots or parametric down-conversion. Compared with the said ways, single-atoms have advantages of narrowband, matching atom transition lines and weak coupling of neutral ground-state atoms with background light and external electromagnetic fields. Single-atom source based on the captured single-atom in an optical tweezer [1–3] paves the way for quantum repeaters, quantum teleportation, quantum secure communications and linear quantum computing. In 1975, Hansch and Schawlow first put forward the use of the laser to cool neutral atoms [4]. In 1987, the research group led by Steven Chu cooled and captured neutral Sodium atoms with magneto-optical trap (MOT) for the first time [5]. In 1994, Kimble's team first achieved the cooling and capture of a single-atom [6]. In 2016, the Browaeys group constructed a

two-dimensional atom array composed of 50 single-atoms trapped in optical tweezers [7]. In 2018, Browaeys group constructed a three-dimensional atom array composed of 72 single-atoms trapped in optical tweezers, optionally manipulating the spatial position of each atom [8]. In our system, we have already captured [2,9] and transferred [10,11] single-atoms to optical tweezer efficiently, built cesium (Cs) magic-wavelength optical dipole trap [2,3], and finally achieved triggered single-photon source at 852 nm based on single-atom manipulation [3,12].

When manipulating the single-atoms in the optical tweezers, it is required that the atoms be captured. We must capture all the atoms in the tweezers before finishing all the operations. Therefore, it is important to prolong the trapping lifetime of atoms in the optical tweezer, to improve the experimental efficiency and the experimental accuracy.

The trapping lifetime of an atom in the optical tweezer is limited by many factors such as recoil heating, laser intensity fluctuation, the degree of atomic vapor, laser beam pointing stability and the residual atom thermal motion. One can suppress the laser intensity fluctuation [13], increase background vacuum degree, improve the laser pointing stability, increase the laser power and adopt the polarization gradient cooling [14] to prolong the trapping lifetime of a single-atom in the optical tweezer. In this work, we discussed the influence of the vacuum degree and the light intensity fluctuation of 1064-nm optical tweezer on atom trapping lifetime and the experimental methods of improving the fluctuation of the laser intensity.

Generally, one can suppress the laser intensity fluctuation with optical mode cleaner [15], optical inject lock [16] or acousto-optical modulator (AOM) feedback [13,17]. Thanks to its flexibility and simplified experimental setup, the AOM feedback becomes the best candidate to suppress the laser intensity fluctuation only by controlling the diffraction efficiency of AOM.

## 2. Single-Atom Magnetic Optical Trap

Ideally, the lifetime of single-atom in the optical tweezer is identical to the lifetime of the atom state, which means the lifetime for a ground-state atom can be extremely long. The trapping life of the ground state atoms in the optical tweezers is limited due to the influence of background atoms collision, the parametric heating of the optical tweezers and the residual thermal motion of the atoms.

The individual atom trapped in the optical tweezer will collide with the free atoms in the Cs gas chamber. In this process the single-atom gains kinetic energy, heats up and eventually escapes from the optical tweezer, resulting in the reduction of the atomic trapping lifetime. In the experiment, we can reduce the collision probability of induced heating by increasing the background vacuum of the cesium atom gas chamber and reducing the number of background atoms. This can extend the trapping lifetime to some extent.

Since the probe of the thermal ionization gauge in the vacuum system is far away from the cesium atom gas chamber, the measured vacuum may be different from the actual vacuum degree in the Cs atomic vapor cell. By measuring the typical time of the MOT, we can know the pressure level in the Cs atomic gas chamber more accurately. In our scheme, due to MOT capturing the atoms in the chamber, it is impossible to isolate trapped atoms from background atoms. Thus, we measure and fit the loading curve instead of using release and recapture to get the typical time of MOT [17]. The background pressure is related to the atomic density $N$ in the vapor cell [18]

$$N = \frac{1}{\tau\sigma} \sqrt{\frac{M}{3k_BT}} \tag{1}$$

where $\tau$ is the atomic trapping lifetime in the MOT, $\sigma$ is the atomic collision cross-section of Cs, $M$ is the mass of Cs atom, $k_B$ is the Boltzmann constant, according to the pressure formula $p = Nk_BT$ the pressure ($p$) in the vapor cell is [18]

$$p = \frac{\sqrt{3Mk_BT}}{3\tau\sigma} \tag{2}$$

Figure 1 shows the loading curve of the magneto-optical trap when the cooling laser power of the MOT is 150 μW, and the magnetic field gradient is 151.41 Gauss/cm. For atom number $N(t = 0) = 0$ as initial condition and fit the curve with $N(t) = N_S\left(1 - e^{-t/\tau}\right)$ [18], the typical time of the MOT is 5.81 ± 0.14 s, and the pressure in the chamber is about $6 \times 10^{-7}$ Pa. The temperature of atoms in the MOT is ~105 μK under similar conditions [19].

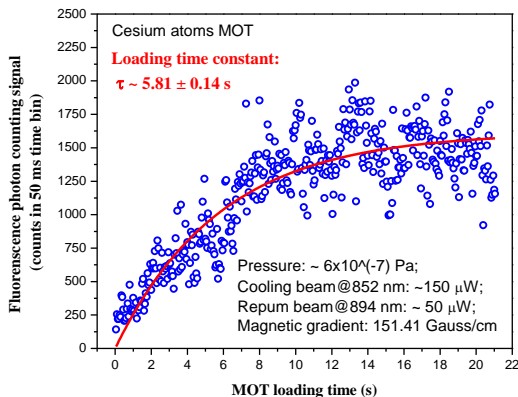

**Figure 1.** Loading curve of the magneto-optical trap (MOT). The MOT is empty when the quadrupole magnetic field is turned off. After the quadrupole magnetic field is turned on, atoms are gradually loaded into the MOT. The typical time constant of the MOT is 5.81 ± 0.14 s, and the corresponding pressure in the Cs vapor cell is approximately $6 \times 10^{-7}$ Pa.

Adjusting the spatial overlap and temporal overlap of the MOT and the optical tweezer can transfer the single-atom between the traps efficiently. The loading rate of the MOT $R_L$ is sensitive to the axial gradient of the quadrupole magnetic field $\left(\frac{dB}{dz}\right)$, and $R_L \propto \left(\frac{dB}{dz}\right)^{\frac{-14}{3}}$ [19,20]. The smaller the axial gradient of the quadrupole magnetic field, the higher the MOT loading rate and the more captured atoms. Conversely, the higher the magnetic field gradient, the lower the MOT loading rate and the fewer atoms trapped.

Since the loading rate of the MOT is pretty sensitive to the axial gradient of quadrupole magnetic field, we can use a trigger loop to control the loading rete of the MOT automatically. As shown in Figure 2a, the fluorescent photons are collected into the single-photon counting module (SPCM), and the output pulses of SPCM enter the pulse counter. Then we can set the output voltage of the pulse counter for different circumstances that represents the atom numbers in the MOT. The output voltage of the pulse counter enters the quadrupole magnetic field power supply (Model SM 70-22, DELTA, Holland) as control voltage. The output current of the power supply will change related to the control voltage, and the quadrupole magnetic field gradient and the loading rate of the MOT change, too.

Figure 2b and Table 1 shows the working principle of the trigger loop. When there is only one atom in the MOT, the output voltage of the pulse counter keeps at 3.8 V, the quadrupole magnetic field gradient stays at 232 Gauss/cm. When there is no atom in the MOT, the output voltage of the pulse counter varies between 1.50 V 1.79 V and let the loading rate of the MOT goes up. Under this circumstance, the output current of the power supply goes down automatically and the quadrupole magnetic field gradient change between 103 Gauss/cm and 116 Gauss/cm. For multi-atom condition, the output voltage of the pulse counter rises, and the loading rate of the MOT descends. On this occasion, the output current of the power supply rises and the quadrupole magnetic field gradient changes between 273 Gauss/cm and 316 Gauss/cm, and the atom number in the MOT cuts down.

Figure 3 shows the probability of the single-atom in the MOT in the cases of trigger loop off or on respectively. The power of the 852-nm MOT cooling laser is 150 μW, and the power of the 894-nm repumping laser is 50 μW. When the trigger loop is off, the loading rate of the single-atom is merely 28.4%, and the 2 atoms loading rate reaches up to 56.8%. The loading rate of single-atom rises to 80.2% when the trigger loop is on, and the multi-atom rate declines sharply.

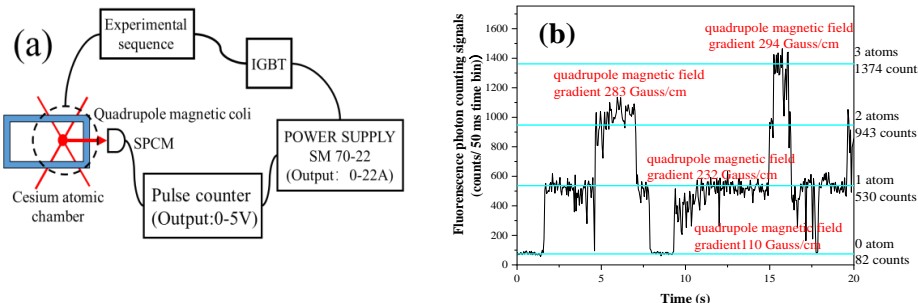

**Figure 2.** Single-atom loading trigger loop. (**a**) Pulse counter adjusts the output voltage according to different photon counts. We use the output analog voltage to control the output current of power supply and the quadrupole magnetic field gradient. Keys to (a) single-photon counting module (SPCM); insulated gate bipolar transistor (IGBT); (**b**) The trigger loop dynamically changes the loading rate of atoms in the MOT by adjusting the quadrupole magnetic field gradient.

**Table 1.** Pulse counter and power supply parameter setting.

| Atom Numbers | Photon Counts (Counts/50 ms) | | Pulse Counter Analog Output Voltage (V) | | Power Supply Output Current (A) | | Quadrupole Magnetic Field Gradient (Gauss/cm) | |
|---|---|---|---|---|---|---|---|---|
| - | Lower Bound | Upper Bound | Lower Bound | Upper Bound | Lower Bound | Upper Bound | Lower Bound | Upper Bound |
| 0 | 0 | 499 | 1.50 | 1.79 | 7.0 | 7.9 | 103 | 116 |
| 1 | 500 | 600 | 3.80 | 3.80 | 17.3 | 17.3 | 232 | 232 |
| 2 | 601 | 1500 | 4.00 | 4.50 | 18.6 | 20.7 | 273 | 304 |
| 3 | 1501 | 2500 | 4.51 | 4.70 | 20.8 | 21.5 | 306 | 316 |

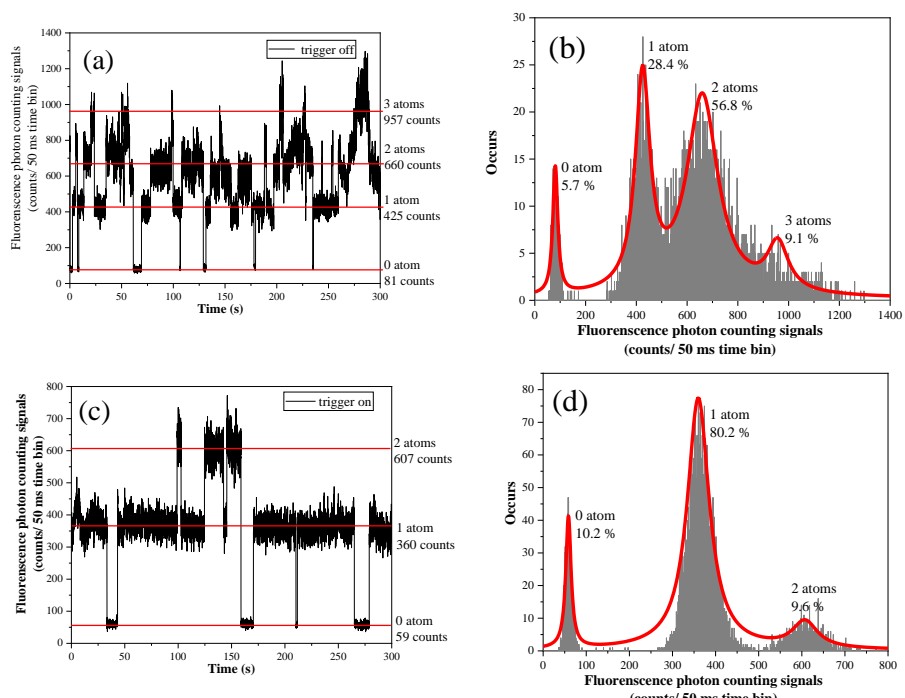

**Figure 3.** The loading probability of single-atom in the MOT. The trigger loop can suppress the probability of multi-atom in the MOT and improve the probability of single-atom. (**a**,**b**) When the trigger loop turned off, the loading probability of single-atom in the MOT within 300 s is only 28.4%, and the probability of multiple-atom and zero-atom is as high as 71.6%, which is obviously not conducive to the measurement in subsequent experiments. (**c**,**d**) For the trigger loop on, the probability of single-atom in the MOT increases to 80.2%, while the probability of multi-atom loading is significantly suppressed.

We can see in Figure 3 that there is still certain photon-counting while there is no atom in the MOT. This is partly because the cooling and repumping laser hits the glass cell wall with scattering and reflecting, the scattered photons at certain specific angles will enter the fluorescence collection system. Moreover, the background atoms move to the MOT laser path will interact with laser and emitted fluorescent photons enter the fluorescence collection system, resulting in the generation of the background photon counting. Besides, since the experimental conditions such as MOT cooling laser power and ambient temperature cannot be the same in each experiment, the background photon count in the MOT cannot be completely consistent but varies within a certain range.

## 3. Parametric Heating of Atoms in an Optical Tweezer

Atoms in optical tweezers will generate electric dipole moments under the action of the oscillating electric field. We can regard these atoms as electric dipoles interacting with the oscillating external field. As shown in Figure 4a when the electric dipole is driven by an electric field with a frequency lower than the resonant frequency of atomic transition, due to the effect of attraction, the atom will be captured in the region with the strongest electric field. Otherwise, when the electric dipole is driven by an electric field with a frequency higher than the resonant frequency, the atom will be excluded from the region with the strongest electric field. In other words, when the frequency of optical tweezer is red-detuning to the atomic transition frequency concerned, the atoms are captured at the bottom of the optical tweezer [21]. When the optical tweezer frequency is blue-detuning to the atomic transition frequency, the atoms will be excluded from the optical tweezer [22].

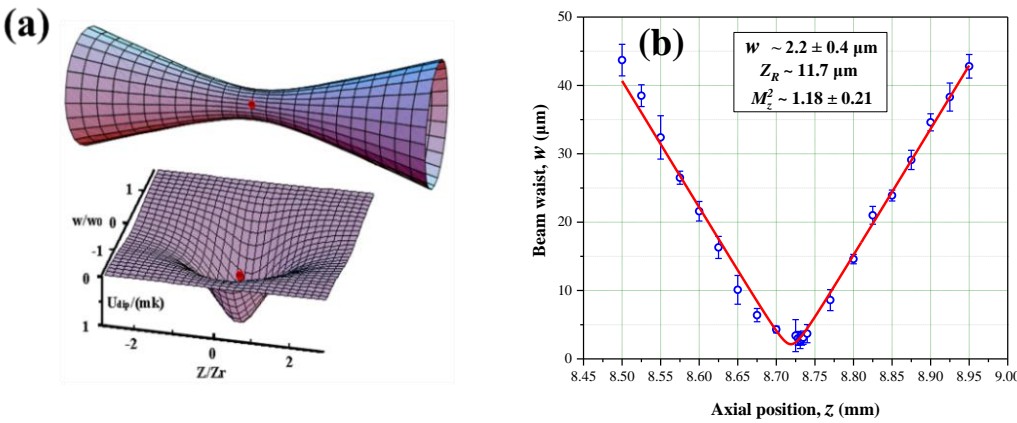

**Figure 4.** Scheme of optical tweezer. (**a**) A strongly focused single-spatial mode Gaussian beam of a red-detuned laser can confine atoms. (**b**) The beam waist radius and Rayleigh length of 1064-nm optical tweezer are 2.2 μm and 11.7 μm, respectively.

Using a focused single-spatial-mode Gaussian beam with red detuning can form an optical tweezer [23], with intensity distribution in the axial and radial direction as follows,

$$I(r,z) = \frac{2P}{\pi w^2(z)} \exp\left[-2\frac{r^2}{w^2(z)}\right] \tag{3}$$

where $P$ is the power of the optical tweezer laser and $w(z)$ is the beam radius at $z$ [23],

$$w(z) = w_0 \sqrt{1 + \frac{z^2}{z_R^2}} \tag{4}$$

for $w_0$ is the beam wrist, $z_R$ is the Rayleigh length, the axial and radial trap frequencies are $\omega_a = \sqrt{\frac{2U}{Mz_R^2}}$ and $\omega_r = \sqrt{\frac{4U}{Mw_0^2}}$, and $U$ is the trap potential depth of the optical tweezer.

The Hamiltonian of single-atom in the optical tweezer is [23]

$$H = \frac{p^2}{2M} + \frac{1}{2} M \omega_{tr}^2 [1 + \varepsilon(t)] x^2, \tag{5}$$

for $M$ is the mass of the Cs atom, $\omega_{tr}^2 = k_0/M$ is the mean square of trap frequency, the elastic coefficient $k_0$ is in direct proportion to the light intensity of optical tweezer $I_0$, we can write the light intensity fluctuation as $\varepsilon(t) = \frac{I(t)-I_0}{I_0}$. Here we use the first-order perturbation theory to clarify how the light intensity fluctuation acts on the single-atom, and take the heating process as the transformation of the atom from $|n\rangle$ at $t = 0$ to $|m \neq n\rangle$ at $t = T$. The average transition probability $R_{m \leftarrow n}$ can express as the power spectral density of the laser intensity fluctuation $S_\varepsilon(\omega)$ [23]

$$\int_0^\infty d\omega S_\varepsilon(\omega) = \int_0^\infty dv S_\varepsilon(v) = \left\langle \varepsilon^2(t) \right\rangle = \varepsilon_0^2, \tag{6}$$

for $\varepsilon_0$ is the mean square of intensity fluctuation and $\omega = 2\pi v$.

The average heating rate of atom in the optical tweezer is [23]

$$\left\langle \dot{E} \right\rangle = \sum_n P(n) 2\hbar \omega_{tr} (R_{n+2 \leftarrow n} - R_{n-2 \leftarrow n}) = \frac{2}{\pi} \omega_{tr}^2 S_\varepsilon(2\omega_{tr}) \langle E \rangle, \tag{7}$$

where $P(n, t)$ is the probability of the atom stay at $|n\rangle$. For $\left\langle \dot{E} \right\rangle = \Gamma_\varepsilon \langle E \rangle$, the constant [23]

$$\Gamma_\varepsilon = \frac{1}{T_{I(\text{sec})}} = \pi^2 v_{tr}^2 S_\varepsilon(2v_{tr}) \tag{8}$$

Among others, $v_{tr}$ is the trap frequency. According to Equation (8), the heating rate of single-atom in the optical tweezer is related to the trap frequency and its double frequency of the optical tweezer. For a single-atom in the optical tweezer, it will not only couple with the fluctuation resonance to the trap frequency of the tweezer, but also couple with the double-frequency more intensely. Ultimately, this heating mechanism makes the atom escape from the optical tweezer. We call the process of converting the energy of the double frequency trap into the energy of the fundamental frequency trap as the parametric process. The heating effect of the intensity fluctuation of the laser on the single-atom in the optical tweezer is the process of parametric heating.

## 4. Measurement of Trap Frequency of Optical Tweezer

The output power of the 1064-nm laser (DBR-1064P, Thorlabs, America) and Ytterbium-doped fiber amplifier (YDFA) system is up to 2 W. The output 1064-nm laser beam passes a lens focal length of 100 mm to form a parallel beam with a diameter of 18–19 mm. Then the parallel beam goes through a lens assembly with a focal length of 36 mm and a numerical aperture of NA = 0.29 to form the optical tweezers. Figure 4b shows the measurement of the beam waist radius and $M_z^2$ factor of 1064 nm optical tweezer are 2.2 ± 0.4 μm and 1.18 ± 0.21, respectively. The Rayleigh length $Z_R$ is 11.7 μm.

As shown in Figure 5, the part in the blue box is used to measuring the trap frequency of the optical tweezer, which is an important parameter of the optical tweezer. We use a function generator (Model DG635 SRS, America) to load modulation signals of different frequencies on AOM 2 (Model 3110-197, Crystal Technology, America) to simulate the intensity fluctuation of the optical tweezer so that the optical tweezer will run with intensity fluctuation at a specific frequency.

Then, by measuring the transfer efficiency of single-atom between the MOT and the optical tweezer to observe how the intensity fluctuation of different frequency acts on single-atom trapping lifetime and Figure 6 shows a typical atomic transfer signal between the MOT and the optical tweezer. As shown in Figure 6, the transfer efficiency gets extremely low when the modulation frequency is resonant with the trap frequency and its double frequency of the optical tweezer.

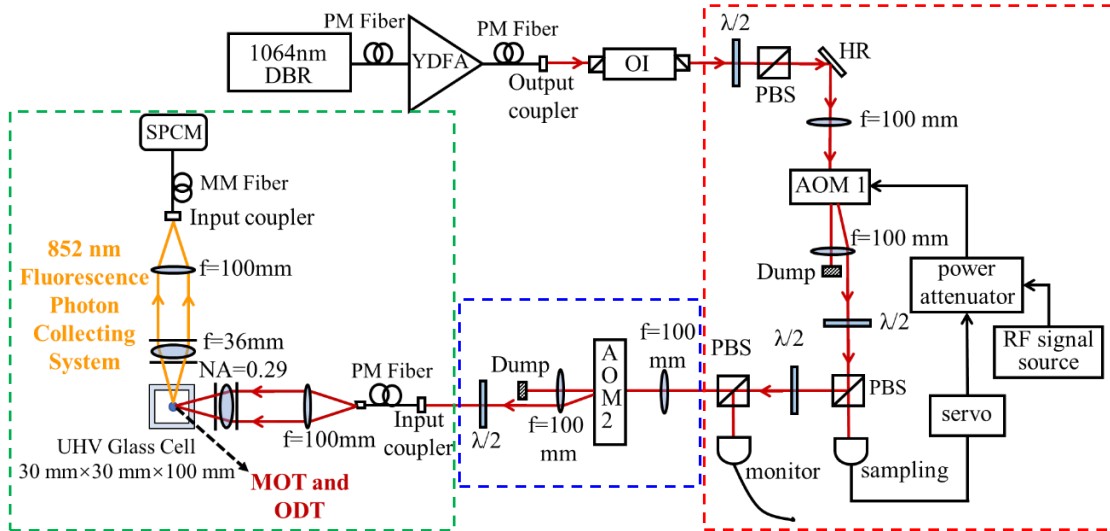

**Figure 5.** Experimental setup. Polarization-maintaining (PM) fiber; Multi-mode (MM) fiber; magneto-optical trap (MOT); optical-dipole trap (ODT), otherwise the optical tweezer; optical isolator (OI); acousto-optical modulator (AOM); polarization beam splitter (PBS); single-photon counting module (SPCM); ultra-high vacuum (UHV). The red dash box shows the laser intensity fluctuation feedback loop by controlling the diffraction efficiency of AOM 1 to stabilize the intensity fluctuation of the 1064-nm laser. The blue dash box shows the measurement setup of the trap frequency of optical tweezer. The green dash box shows the single-atom cooling and trapping system, the anti-Helmholtz coils, but the cooling and repumping laser beams of the MOT are not shown here.

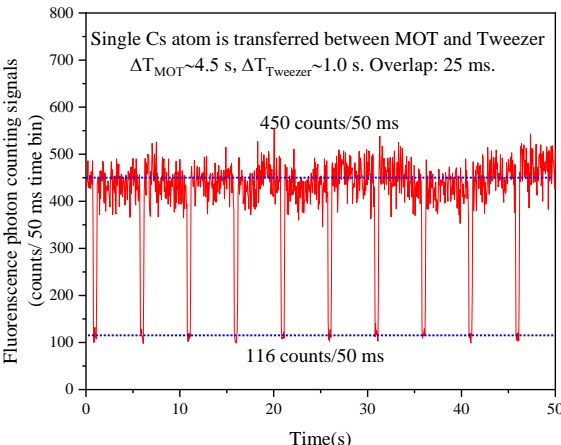

**Figure 6.** Atomic transfer signal between the MOT and the optical tweezer. The period is 5.5 s, in which the MOT duration is 4.5 s and the optical tweezer last for 1.0 s. The MOT overlaps with the optical tweezer for 25 ms, and the single-atom is transferred between the MOT and the optical tweezer.

Figure 7 shows the trap frequency and its double frequency on the axial and radial direction. The power of 1064 nm optical tweezer is 48.5 mW, the optical tweezer trap depth is 1.3 mK. The measured axial trap frequency is 4.70 kHz, for double frequency is 9.00 kHz (should be 9.40 kHz), the radial trap frequency is 41.80 kHz, and the double frequency is 80.80 kHz (should be 83.60 kHz). The deviation between the experimental value and the theoretical calculation is mainly because that the suppression effect of the laser intensity fluctuation feedback loop cannot be kept at the same level due to the long duration of the experimental process.

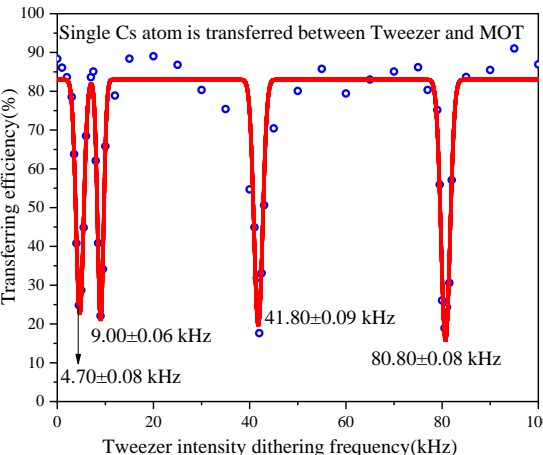

**Figure 7.** Trap frequency and double frequency of the 1064-nm optical tweezer. Transfer efficiency goes an apparent decline when the modulation frequency is resonant or near-resonant to the trap frequency and its double frequency of the optical tweezer and reaches a minimum at resonance.

The photon scattering rate of the optical tweezer is [24]

$$\Gamma_{SC} = \frac{3\pi c^2 \omega_L^3}{2\hbar\omega_0^6} \left( \frac{\Gamma}{\omega_0^2 - \omega_L^2} + \frac{\Gamma}{\omega_0^2 + \omega_L^2} \right)^2,\tag{9}$$

where $\Gamma$ is the spontaneous decay rate of the atomic transition, $\omega_0$ is the transition frequency, $\omega_L$ is the frequency of the optical tweezer. In a 1064-nm optical tweezer, for Cs 6 $S_{1/2}$ ($F_g = 4$) −6 $P_{3/2}$ ($F_e = 5$) transition, the photon scattering rate is 8.3 photons/s.

## 5. The Suppression of Laser Intensity Fluctuation

### 5.1. Experimental Principles and Devices

Figure 5 shows the scheme of the laser intensity fluctuation control system. The part in the green box is the single-atom capturing and observing system. The optical fiber output beam goes through the collimating lens and expended to a near-parallel beam with a beam diameter of 19 mm. Then the parallel beam passing through the focusing lens assembly (NA = 0.29), and become a tightly focused Gaussian beam with a beam wrist of 2.2 µm. We use the same lens assembly to collect the fluorescence photons of Cs atoms and transfer them into the SPCM through an optical fiber.

The part in the red box is the laser intensity fluctuation feedback loop. The first-order diffraction light of AOM 1 as the optical tweezer laser, the zero-order diffraction light as the "energy storehouse" of the first-order diffraction light. The first-order diffraction beam goes through the PBS, the transmission light as the sampling laser, and collected into the detector (New Focus 2051), and the laser intensity fluctuation reflected as the detector's output voltage. Then the output voltage signal enters the Proportion Integration Differentiation (PID) (Model SIM960, SRS, America) as the reference signal and set feedback-signal-related parameters (e.g., integral time, etc.) of the PID. Next, load the feedback signal on the driving voltage of AOM 1, we can control the diffraction efficiency of AOM 1 precisely. When the optical tweezer laser (the first-order of the diffracted light of AOM 1) intensity decreases (or increases), the output voltage of the sampling detector decreases (or increases) too. The PID output positive feedback (or negative feedback) signal will make the diffraction efficiency of AOM 1increases (or decreases), which means the power of the first-order diffraction light goes up (or down), and of zero-order diffraction light goes down (or up). Finally, we can achieve the purpose of suppressing the fluctuation of laser intensity.

### 5.2. The Suppression Results of Laser Intensity Fluctuation

Figures 8 and 9 show the response of the feedback loop. The total power of the 1064-nm laser is 1.6 W and the sampling power is 4.1 mW. The light intensity fluctuation in the time domain for the free-running case and feedback-on case is ±3.360% and ±0.064%, respectively. The feedback bandwidth in the frequency domain is 33 kHz, which covers the axial trap frequency and its double frequency of optical tweezer, and is much broader than that in our previous works [13,17]. There is a strong fluctuation peak on 19.1 kHz, due to the fluctuation from the external environment. The other four fluctuation peaks on 5.9 kHz and its double frequency (11.8 kHz), triple efficiency (17.7 kHz) and quadruple frequency (23.6 kHz) are caused by YDFA's cooling system.

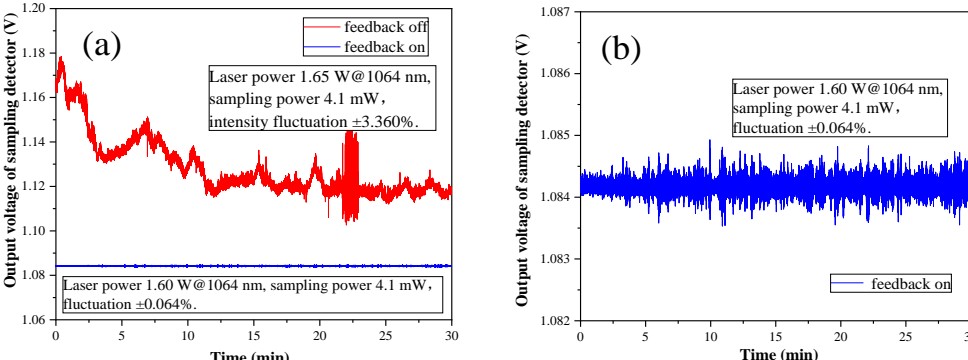

**Figure 8.** Intensity fluctuation of laser in the time domain in the free-running case and the feedback-on case. The light intensity is suppressed from ±3.360% to ±0.064%: 1.6 W for a total power of the 1064-nm laser, and the sampling power is 4.1 mW (**b**), zooming of the feedback-on case in (**a**).

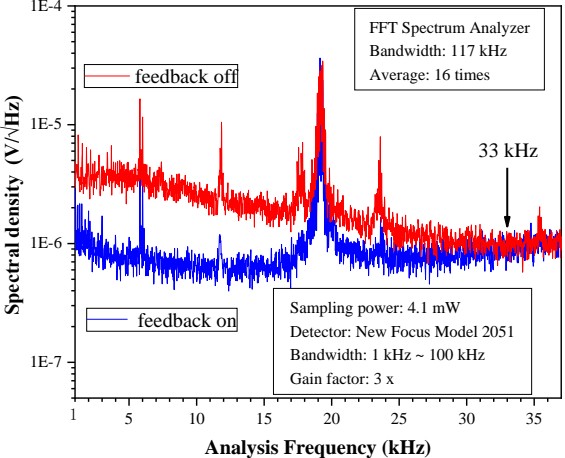

**Figure 9.** Comparison of the intensity fluctuation of laser in the frequency domain when feedback loop is on and off. The feedback bandwidth in the frequency domain is 33 kHz. Several noise peaks are occurring from the external environment or the YDFA's cooling system.

### 6. Improvement of Atom Trapping Lifetime in Optical Tweezers

In the experiment, the power of the 852-nm cooling laser is 105 μw, the power of the 894-nm repumping laser is 50 μW, the quadrupole magnetic field intensity is 254 Gauss/cm, and the power of the 1064-nm optical tweezers is 50 mW. Firstly, we capture a single-atom in the MOT and adjust the spatial overlap of the MOT and the optical tweezer. We change the experimental sequence to control the time overlap of the optical tweezer and the MOT for 25 ms, which maximizes the atomic transfer efficiency between the MOT and the optical tweezer. Next, we measure the single-atom transfer efficiency under different time duration of the optical tweezer. As the duration of optical tweezer

increases, atom transfer efficiency goes down exponentially. Finally, the single-atom trapping lifetime in the 1064-nm optical tweezer can be obtained by fitting measured data.

As shown in Figure 10, the trapping lifetime of single-atom in the 1064-nm optical tweezer without the feedback loop on is only about 4.04 ± 0.92 s, while the trapping lifetime is 6.34 ± 0.41 s after the feedback loop is on. We can see that the trapping lifetime of single-atom in the optical tweezers is extended after the suppression of the intensity fluctuation of the optical tweezer.

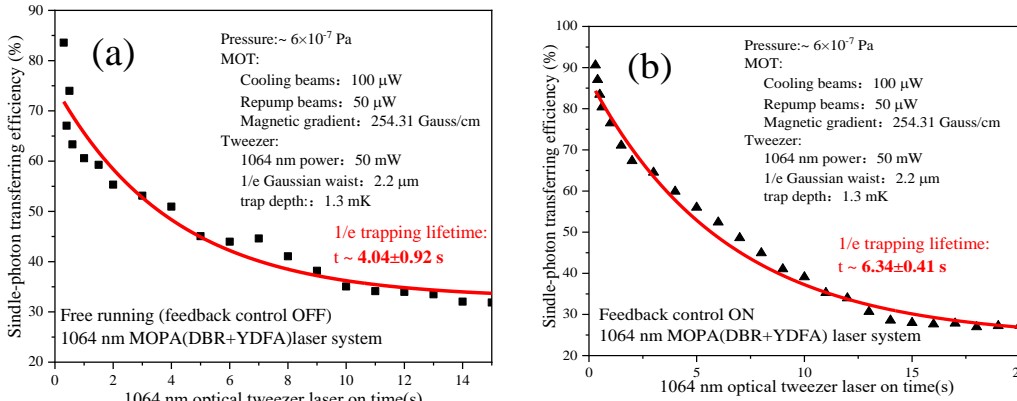

**Figure 10.** Trapping lifetime of atom in 1064-nm optical tweezer when the feedback loop is turned off (**a**) and on (**b**). The trapping lifetime of single-atom is extended from 4.04 s to 6.34 s.

The typical trapping lifetime here we achieved seems shorter than that of our previous work [19] because the background Cs density in the ultra-high vacuum (UHV) glass cell has increased a lot in this experiment. Now the trapping lifetime of single-Cs atom in the optical tweezer is mainly limited by the atomic collisions under the typical pressure of approximately $6 \times 10^{-7}$ Pa.

## 7. Conclusions

In this paper, we analyzed how the background pressure in the atomic vapor cell and the intensity fluctuation of optical tweezers act on the atomic trapping lifetime and suppressed the intensity fluctuation of the 1064-nm optical tweezer to extend the trapping lifetime of single-atom. The suppression bandwidth will be extended to cover the trap frequency and its double frequency on the both axial and radial direction. In addition, the suppression effect of laser intensity fluctuation in the time domain and the feedback bandwidth in the frequency domain can be adjusted to meet different experimental requirements, which would provide valuable insights for subsequent experiments of single-atom manipulation and quantum simulation.

**Author Contributions:** The experiment and data were completed by R.S., X.W. and K.Z.; J.H. undertook partial guidance during the experiment. And J.W.'s contributions included coordination, guidance and data analysis in the experiment; R.S. and J.W. wrote and revised the manuscript. All authors have read and agreed to the published version of the manuscript.

**Funding:** This research was funded by [The National Key R & D Program of China] grant number [2017YFA0304502], [the National Natural Science Foundation of China] grant number [11774210, 6187511, 11974226 and 61905133], and [Outstanding Graduate Innovation Program of Shanxi Province] grant number [2019BY2016].

**Conflicts of Interest:** There is no conflict of interest in this article.

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
