# Peer review of "Influence of Laser Intensity Fluctuation on Single-Cesium Atom Trapping Lifetime in a 1064-nm Microscopic Optical Tweezer"

_applsci, doi:10.3390/app10020659_

Round 1

Reviewer 1 Report

Equation 1, the m is not defined. Line #157, is the M the same as that in Eq. (5)? If so, then U is in the energy unit. It is not the depth of the optical tweezer. Figure 7 and its description are not clear. The authors need to mark clearly the parts and their functions. The authors present the theories in  Eq. (9) and Eq. (13), but the experiment and the conclusion seem to have no relation with the theories.  

Reviewer 2 Report

This paper examines ways of prolonging the trapping lifetimes of atoms in an optical tweezer. The described process results in improved experiment efficiency and accuracy. The paper begins with a good background history of single atoms as single photon sources. References are plentiful. The review includes a summary of the authors' past research and publications regarding optical tweezers. The authors analyze the influences of the background vacuum, including the role of neighboring atoms, the trap frequency of optical, and the laser intensity fluctuations. Using an external feedback loop to reduce intensity noise from 3.360% to 0.064%, the authors extend capture time from 4.04 s to 6.34 s. The experimental process is well documented. It is detailed with sufficient graphs and diagrams. The theoretical basis is well developed. The paper is clearly and concisely written. The findings are important for the field. The paper is very deserving of publication.

Author Response

Response to Reviewer 2 Comments

Point: This paper examines ways of prolonging the trapping lifetimes of atoms in an optical tweezer. The described process results in improved experiment efficiency and accuracy. The paper begins with a good background history of single atoms as single photon sources. References are plentiful. The review includes a summary of the authors' past research and publications regarding optical tweezers. The authors analyze the influences of the background vacuum, including the role of neighboring atoms, the trap frequency of optical, and the laser intensity fluctuations. Using an external feedback loop to reduce intensity noise from 3.360% to 0.064%, the authors extend capture time from 4.04 s to 6.34 s. The experimental process is well documented. It is detailed with sufficient graphs and diagrams. The theoretical basis is well developed. The paper is clearly and concisely written. The findings are important for the field. The paper is very deserving of publication.

Response: Thanks for the reviewer’s encouragement to our works.

Reviewer 3 Report

See attached

Round 2

Reviewer 3 Report

Thank you for your careful attention to my comments, and for highlighting your changes.  With one minor correction, and some editing of the English, the paper will be suitable for publication.

I still cannot find equations 1 and 2 in reference 18, attached

[18] Grimm a., Weidemüller M., and Oechinnikoe Y. B., “Optical dipole traps for neetral atoms”, Adv. At. Mol. 357 Opt. Phys., 2000, 42, pp 95–170. 

I am sure it is simple, but the equations need to be properly referenced. 

It would help if you explicitly stated that the Rayleigh range is adjusted for M^2.
